# Pore Segmentation Techniques for Low-Resolution Data: Application to the Neutron Tomography Data of Cement Materials

**DOI:** 10.3390/jimaging8090242

**Published:** 2022-09-07

**Authors:** Ivan Zel, Murat Kenessarin, Sergey Kichanov, Kuanysh Nazarov, Maria Bǎlǎșoiu, Denis Kozlenko

**Affiliations:** 1Frank Laboratory of Neutron Physics, Joint Institute for Nuclear Research, 141980 Dubna, Russia; 2Laboratory of Neutron Physics, The Institute of Nuclear Physics Ministry of Energy of the Republic of Kazakhstan, Almaty 050032, Kazakhstan

**Keywords:** pore segmentation, watershed, enhanced contrast, neutron tomography, MKP cement

## Abstract

The development of neutron imaging facilities provides a growing range of applications in different research fields. The significance of the obtained structural information, among others, depends on the reliability of phase segmentation. We focused on the problem of pore segmentation in low-resolution images and tomography data, taking into consideration possible image corruption in the neutron tomography experiment. Two pore segmentation techniques are proposed. They are the binarization of the enhanced contrast data using the global threshold, and the segmentation using the modified watershed technique—local threshold by watershed. The proposed techniques were compared with a conventional marker-based watershed on the test images simulating low-quality tomography data and on the neutron tomography data of the samples of magnesium potassium phosphate cement (MKP). The obtained results demonstrate the advantages of the proposed techniques over the conventional watershed-based approach.

## 1. Introduction

Over the past years, neutron imaging (neutron radiography and tomography) has become a widely used non-destructive method realized at dozens of neutron facilities over the world [1]. The specifics of neutron interaction with matter, such as high penetration into material volume and sensitivity to both light and heavy nucleus, provoke a growing range of applications of neutron radiography and tomography [2,3] including engineering, plant science, petrophysics, cultural heritage, cement research, etc. In the particular case of cement materials, aggregates, cracks, pores and other kinds of inclusions or phases inside the cement matrix can be visualized using neutron tomography [4]. Based on the obtained structural data, the physical and mechanical properties of cement materials can be predicted [5].

Despite unique and advantageous of the neutron imaging method in comparison with other imaging techniques (e.g., X-ray CT), there are technical difficulties in a high-quality measurement setup development associated with a beam collimation, efficient detector system, low radiation background, and the desired energy spectrum of neutrons.

The resolution of the neutron images is mostly constrained by the parameters of the pin-hole geometry, thickness, and efficiency of the scintillator screen [6]. Complex radiation conditions and background, which are formed by scattered neutrons from the sample itself, as well as an unfocused camera, additionally decrease the image quality. As an example, schematically shown in Figure 1 is the formation of a neutron radiographic image of the phantom ‘hole’ that simulates a pore in a material. The hole is transparent for neutrons, i.e., the neutron attenuation coefficient equals zero everywhere within its boundaries. However, we find that the intensity of neutrons behind the phantom after the flat field correction is not uniform, and depends on the distance between the phantom and the scintillator screen. Therefore, even the center point of the hole may appear to be attenuative (see Figure 1). The center point of the hole will be fully transparent for neutrons (incoming intensity equals detected one) if the following condition is fulfilled:(1)$$d_{h}\geq l\frac{1}{L/D}$$
where *d_h_*—hole diameter of the phantom, *l*—distance from the phantom to the scintillator screen, *L*—distance between the pin-hole and the scintillator screen, and *D*—diameter of the pin-hole. Due to *l* ≠ 0, the resulting radiographic image of the phantom is blurred, and the holes with different sizes will appear differently in the image following Equation (1). The presence of the additional background brings corresponding artifacts to the data, making the studied sample artificially less attenuative [6,7]. Assuming the intensity of the background to be small compared to the intensity of the neutron beam, the contribution of the background to the measured projection may be expressed through:(2)$$P_{meas}\approx P_{sample}-e^{P_{sample}}\frac{\Delta I_{background}}{I_{0}},$$
where *P_meas_*—measured projection, *P_sample_*—projection solely of the sample, ∆*I_background_*—background intensity, and *I*_0_—open beam intensity. Equation (2) must be modified to account for the additional blur coming from the scintillator and unfocused camera. As a result, measured neutron images can be expressed as
(3)$$I_{measured}=B_{camera}*B_{scint}*\left(I_{sample}+\Delta I_{background}\right),$$
where ∗ denotes the convolution, and $B_{camera},\;B_{scint}$ are the kernels defining the blur from unfocused camera and scintillator, respectively.

The considered example of the phantom ‘hole’ is intended to demonstrate the formation of neutron images of porous materials and to highlight the fact that the data obtained after the tomographic reconstruction may have a low resolution and artifacts. The arising problem is how to discriminate the pores against other phases in the reconstructed images, because the aforementioned corruptions produce the effect of uneven illumination making some pores more or less attenuative in neutron images than others.

There is a large number of segmentation methods and their modifications used in the image analysis in different fields of research. Concerning pore segmentation in the tomography data, Otsu’s thresholding [8], k-means [9], region growing [10], watershed [11], kriging [12], and machine learning techniques [13] have been commonly utilized. The extensive overview by [14,15] covers most of the segmentation techniques used in X-ray tomography studies of porous materials. However, there is no any universal pore segmentation technique that robustly works for any kind of data, especially, when the image quality is poor [16,17]. In addition, we have not found in the literature the pore segmentation techniques specifically designed for neutron tomography data.

In this work, we present pore segmentation techniques for low-resolution data. One of them is the application of the morphological enhanced contrast operators, while the second one is the extension of the conventional watershed-based technique, which we called local threshold by watershed segmentation (LTWS). The proposed techniques were compared on the test images simulating low-quality tomography data and on the ‘real’ neutron tomography data obtained for the samples of magnesium potassium phosphate (MKP) cement.

## 2. Pore Segmentation Techniques

### 2.1. Global Threshold of the Enhanced Contrast Data

The quality of the reconstructed data significantly suffers from blurring effects and artifacts. The corresponding effect of the uneven illumination does not allow for the reliable choice of the global threshold. At any chosen gray level, as a global threshold, there will be the segmented pores, which are oversized, undersized, or both, as compared to the ground truth. The effect of the uneven illumination may be diminished by enhancing the overall contrast of the image, i.e., by increasing the gradients at the boundaries of phases. Then, the corrected image can be binarized using the global threshold.

The corresponding enhanced contrast operator for the gray-tone image *f* can be constructed from the morphological top-hat operators [18]. The top-hat contrast operator for a given structural element or a connected neighborhood is defined as:(4)$$\kappa^{TH}\left(f\right)=3f-\phi \left(f\right)-\gamma \left(f\right),$$
where ϕ and γ are the morphological closing and opening operators, respectively. The output image is further constrained to the dynamic range of an input image as $\left[0,\;f_{max}\right]$ in case of porous materials. Subsequent application of $\kappa^{TH}$ multiple times will finally transform the input image into the binary one. However, the result of such ultimate binarization will show connected regions of local maxima and minima, rather than the segmented regions corresponding to pores. This is also due to the fact that most of the studied materials are presented by more than two phases (e.g., pores, matrix, and highly attenuative phase inside the matrix), and thus require at least trinarization.

The dynamic range of the top-hat contrast operator can be saturated by adding factor *n*, which scales the impact of the top-hat operators. By such definition:(5)$$\kappa_{n}^{TH}\left(f\right)=\left(2n+1\right)f-n\cdot \phi \left(f\right)-n\cdot \gamma \left(f\right),$$
and the contrast operator can be tuned by *n* as any positive number. A range of *n* < 1 is useful for preserving the integer format of the output image or for constraining the effect of the contrast operator. We note that there is finite number $n_{\max}$, at which the top-hat contrast operator (5) becomes independent on *n,* and the output image becomes *flattened*, so that the local minima or maxima no longer exist (but only global extrema).

The enhanced contrast operator can also be constructed from the median filter as well. If *m* is medium operator, then the following decomposition into *increasing* ($m_{i}$) and *decreasing* ($m_{d}$) medium filters can be used:(6)$$m_{i}=m\vee id=\max\left(m,id\right);\;m_{d}=m\wedge id=\min\left(m,id\right),$$
where id is an identity operator. From this definition, it follows:$$m=m_{i}+m_{d}-id.$$

While the medium filter is a self-dual operator with respect to the complementation C: CmC=m; its increasing and decreasing versions are dual: $Cm_{d}C=m_{i}$. Although medium filters are not idempotent, they share some common properties with morphological opening and closing. Thus, the alternative contrast operator can be introduced, which is a self-dual one:(7)$$\kappa^{m}\left(f\right)=3f-m_{i}\left(f\right)-m_{d}\left(f\right)=2f-m\left(f\right),$$

The corresponding extended version $\kappa_{n}^{m}\left(f\right)$ is constructed in the same way as for the top-hat contrast operator in Equation (5):(8)$$\kappa_{n}^{m}\left(f\right)=\left(2n+1\right)f-n\cdot m_{i}\left(f\right)-n\cdot m_{d}\left(f\right)=\left(n+1\right)f-n\cdot m\left(f\right).$$

According to Equations (7) and (8), medium-based enhanced contrast operators can be computed without a decomposition (6). Also, Equation (8) may be further extended for the use of other filters, e.g., mean and gaussian. If we denote by *s* such a smoothing operator, then the corresponding enhanced contrast operator can be written:(9)$$\kappa_{n}^{s}\left(f\right)=\left(n+1\right)f-n\cdot s\left(f\right).\;$$

The obvious advantage of using the medium filter in (9) is its edge-preserving property, which is valid at a low signal-to-noise ratio [19].

The size of either the window or the structural element for contrast operators depends on whether large or small regions should be sharpened in the image. Finally, images with enhanced contrast can be binarized using the global threshold, since the illumination variations, which are larger than the window size of the contrast operator, became smaller with respect to the difference between the bright and dark regions in the image.

### 2.2. Watershed-Based Techniques

#### 2.2.1. Conventional Approach (WS)

The purpose of the segmentation is to determine the boundaries between phases. For the gray-tone image, the boundary can be chosen as the locus of points with the highest gradient. The problem of the segmentation of phase boundaries employing the gradient image is efficiently solved by the watershed transform (WS) [20]. The gradient image may be visualized, as the topographic landscape with ridges as local maximum, and valleys as local minima. The watershed transform decomposes this image, showing only the catchment basins of all valleys, which are separated by the watersheds. By computing the watersheds, the seeking boundaries between phases can be found. Since we are only interested in pores, an additional image is required—marker image. The marker image is a binary image showing the approximate locations of pores in the original image. It can be thought of as an image of seeds placed at the positions of pores. With a help of the marker image, we can eliminate all local minima in the gradient image, which do not correspond to the pores by using the minima imposition technique [18]. In addition, this operation suppresses spurious local minima and prevents over segmentation. In the present work, we use the following sequencing: gauss filtering of the original image, calculation of the gradient image, minima imposition using marker image, and watershed transform.

#### 2.2.2. Local Threshold by Watershed (LTWS)

The conventional WS approach fully relies on the gradient image. However, the presence of false local maxima in the gradient image, due to a noise and blur, will lead to false segmentation. In general, the image intensity along the watershed contours varies and does not correspond to the single gray level. Based on the fact that each pore can be segmented by its own threshold, we propose the following modification of the conventional watershed approach employing the gradient image. The idea of the proposed method is to compute the local threshold for each of the pores, which were segmented by the watershed. The value of the threshold is found as the minimum among the gray values of the original or filtered image belonging to the corresponding watershed lines. This operation helps to prevent or at least to minimize the false segmentation appearing due to the presence of wrong maxima in the gradient image. We propose to add the following steps to the conventional WS method according to the proposed LTWS: labeling of pores segmented by the conventional WS, for each labeled pore (catchment basin) compute the threshold as the minimum value of the original image at the corresponding watershed line or simply at the pore’s boundary; binarization of each labeled pore using the calculated threshold, and final compilation of all segmented pores in one binary image (or 3D binary data).

## 3. The Segmentation Test

We have tested the presented segmentation techniques on the set of artificial images with different quality. The original (gray-tone ground truth) image represents the distribution of four different phases (Figure 2). One of them with zero intensity corresponds to pores. We have used two different subsets: one is with a flat background, and another one with a parabolic background shape, which has a minimum at the image center. All images were corrupted by the blurring and noise, using built-in plugins of the ImageJ [21]. The following variants of image corruptions were used. The first variant coded as s4_n_s2 represents the corruption of the original gray-tone image by Gaussian blurring with the sigma parameter of 4, Gaussian noise with a mean of 0, and standard deviation of 75, and Gaussian blurring with the sigma parameter of 2. The second variant coded as s6_n_s2_n_s2 was obtained by Gaussian blurring with the sigma parameter of 6, twice the application of the sequence of Gaussian noise with a mean of 0, and standard deviation of 75, and Gaussian blurring with the sigma parameter of 2.

Pore segmentation of the test images was performed using marker-based WS, the proposed extension LTWS and medium-based (window of 150 pixels) enhanced contrast image, which was thresholded at the minimum gray level of $\kappa^{m}\left(f\right)=0$. Marker images were obtained using the global threshold *t*. We have slightly modified the conventional WS by changing the output to $\max\left(\mathrm{WS}\;,marker\;image\right)$. Such modification are necessary when the marker image overstep the corresponding watershed line; otherwise, the WS fails. We denoted the WS without this modification, as WS*.

The results of pore segmentation in the tested images are shown in Figure 2. A quantitative assessment of the difference between the ground truth and the segmented images was performed using the Jaccard index [22] (see Table 1), defined as $J\left(A,B\right)=\frac{\left|A\cap B\right|}{\left|A\cup B\right|}$, where *A* and *B* are the binary images. The Jaccard index is a well-known measure of the similarity of two finite sets. Using a definition, $0\leq J\left(A,B\right)\leq 1$, where 0 means that *A* and *B* have no matches and 1 means a perfect match. According to Figure 2 and Table 1, the proposed extension LTWS showed itself as being a more reliable technique than the conventional WS. Independently, on the input image quality or the background, the LTWS tends to preserve the shape of the pores, while the conventional WS provides more oversized pores with a tendency to pore-shape deformation. Moreover, the quality of the WS strongly depends on the marker image, i.e., on the threshold level that we have chosen to binarize the input image. In turn, the LTWS shows stable results for all marker images, and did not fail even for the worst quality image of our set. However, the best Jaccard index in most cases was obtained for the images binarized using the global threshold of the enhanced contrast image (Table 1). As seen in Table 1, this technique is close to the best results obtained using the watershed-based methods. It also tends to preserve the shape of pores and is almost independent of the quality of the test images.

We note that the conventional watershed without our modification WS* already failed on the first test image (Figure 2). This happened because the operation of minima imposition using the marker image that oversteps the actual watershed line suppressed the corresponding local maxima in the gradient image.

## 4. Application to the Neutron Tomography of MKP Cements

### 4.1. Experimental

Four magnesium potassium phosphate MKP cement samples were studied using the neutron tomography method. All samples were prepared following exactly the same formulation: MgO (10 g) + KH_2_PO_4_ (35 g) + Fly ash (45 g) + 2% Boric acide (0.9 g) + 2% LiNO_3_ (0.9 g) + Sand (45 g) + 17 mL D_2_O. There were labeled as MKP_1, MKP_2, and MKP+Al_1, MKP+Al_2, which additionally contain 1.75 g of aluminum platelets. The samples have a parallelepiped shape with a height of about 5 cm and a width of 1 cm. More information about cement sample preparation and chemical aspects were presented previously [23].

The neutron tomography experiments were performed at the neutron radiography and tomography facility placed on beamline 14 of the IBR-2 high-flux pulsed reactor [24]. The neutron flux at the sample position is ~5.5 × 10^6^ n × cm^−2^ × s^−1^. A set of neutron radiography images was collected by the detector system based on a high-sensitivity camera with a HAMAMATSU CCD chip (2048 × 2048 pixels). A field of view of 10.5 × 10.5 cm^2^ was used for MKP_1 and MKP_2 samples, while for MKP+Al_1 and MKP+Al_2, a larger field of view of 12 × 12 cm^2^ was used. The neutron tomography experiments were performed with a rotation step of 0.5°, corresponding to the 360 measured radiography projections. The exposure time for one projection was 20 s, image acquisition and sample rotation took additionally about 20 s per image, so the resulting measurements lasted for about 4 h for each of the cement samples. The distance between the center of rotation of the sample and the scintillator screen was about 60 mm. The spatial resolution capabilities of the neutron tomography facility have some restrictions on the minimum size of a resolved item up to ~135 µm. The imaging data were noise filtered and then corrected by the camera dark current image and normalized to the image of the incident neutron beam. Stripes removal and tomographic reconstruction were performed using SYRMEP Tomo Project software [25]. In particular, the wavelet-Fourier filtering technique [26] was used for sinogram filtering, and tomographic reconstruction was performed using the simultaneous algebraic reconstruction technique [27].

Virtual 3D models of cement samples obtained from tomographic reconstruction (Figure 3) depict the spatial distribution of the neutron attenuation coefficient expressed in cm^−1^ units. The observed uneven distribution of the intensity in the gray-tone data is related to the presence of phases with different neutron attenuation coefficients. The highest attenuation corresponds to the regions enriched with B, H, and Li elements with high absorption or scattering cross-sections of thermal neutrons [28]. The regions with the lowest attenuation corresponded to the pores. In the slices shown in Figure 3, the complex distribution of the attenuation coefficient depicting different phases including pores and products of chemical reactions during the cement hardening is clearly seen. However, the calculated histograms of the attenuation coefficient over the 3D data sets have two major peaks (Figure 3). One of them is the background and pores, while the second peak is the cement matrix and other solid phases. The large valley of the non-zero values between them corresponds to the smooth boundaries (with relatively small gradients) between phases in the reconstructed data.

### 4.2. Pore Segmentation Results

We performed the segmentation of neutron images using the same techniques used in the segmentation tests (see Section 3). They are the global thresholding, the medium-based (window of 100 pixels) enhanced contrast image thresholded at the minimum gray level ($\kappa^{m}\left(f\right)=0$), the conventional marker-based WS, and the proposed extension LTWS. We used the same threshold for both the conventional global threshold technique and for the marker image. However, the choice of the global threshold was not based on the histogram data (Figure 3). For this purpose, we performed the WS technique in 2D slices to the sample itself and found its boundaries. Following the idea of LTWS, we collected the minimum gray values at the watersheds for each of the 2D slices. The resulting histogram is shown in Figure 4. We obtained almost the same Gaussian-like distribution for all samples. Trying to preserve the maximum number of pores, we chose the global threshold as the maximum value over the obtained distributions among all samples.

In Figure 5, binary images with the segmented pores obtained by different techniques are shown. For all samples, we can see noticeable differences between conventional WS and all other techniques. In the case of the MKP_2 sample, even WS has failed and shows obvious spurious segmentation. As was expected, the LTWS method provided more reliable results. In comparison with the results of the segmentation tests (Figure 2), the difference between the binarized enhanced contrast data and the LTWS is somewhat larger.

The calculated porosities based on the obtained segmentations are listed in Table 2. It is evident that WS provides drastically different values than other methods, which may be explained by the wrong segmentation. However, the enhanced contrast technique and the conventional global thresholding demonstrated almost the same results for the samples’ porosity (Table 2), while the application of LTWS yielded almost the same porosity for all samples of about 0.4%.

Enhanced contrast operators (Equations (5) and (7)) help to increase the phase-to-phase contrast and even can be used for unsupervised binarization. However, we cannot declare the robustness of the corresponding segmentation results, because image binarization by using the global threshold cannot provide segmentation without the potential biases from the user’s choice or from the low quality of the image. Scale parameter *n* (Equation (5)) brings more variability into image processing, but does not resolve the ambiguity problem of the parameter choice. In contrast, the proposed LTWS technique relies only on the marker image and takes into account the information from both the input grey-tone image and its gradient version. The results of tests (Figure 2) and application on the neutron images (Figure 5) have shown encouraging results and the potential of this technique for pore segmentation in low-quality data.

## 5. Conclusions

We have presented new techniques for pore segmentation in low-resolution images or tomography data. They are the global thresholding of the enhanced contrast data and the local threshold by the watershed (LTWS). The performed tests demonstrated their possibilities and also their advantages over the conventional marker-based watershed technique. The considered techniques were applied to the neutron tomography data of the MKP cement samples. The following comparison of the segmented data, as well as of the calculated porosity of the cement samples, confirmed the results of the tests, showing the failings of the conventional watershed as compared to the proposed techniques.

## Figures and Tables

**Figure 1 jimaging-08-00242-f001:**
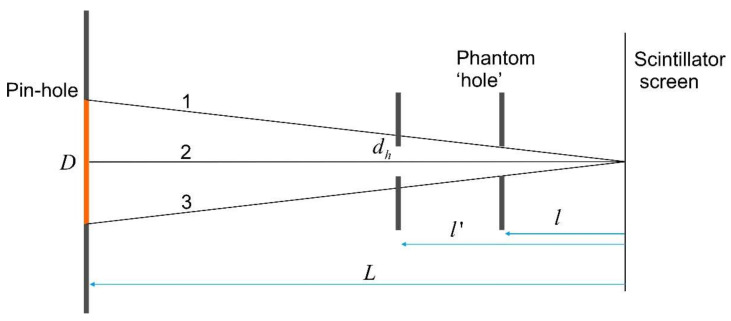
Illustration of the blurring effect in the pin-hole geometry (see Equation (1)). Numbers denote three base rays of the neutron beam. When the phantom is placed at distance *l* from the scintillator the center point of the hole will be absolutely transparent for neutrons, because $\frac{I}{I_{0}}=\frac{I_{1}+I_{2}+I_{3}}{I_{1}+I_{2}+I_{3}}=1$. At distance $l^{\prime }$ $\frac{I}{I_{0}}=\frac{I{’}_{1}+I_{2}+I{’}_{3}}{I_{1}+I_{2}+I_{3}}<1$, and the hole appears to be attenuative even in its center point.

**Figure 2 jimaging-08-00242-f002:**
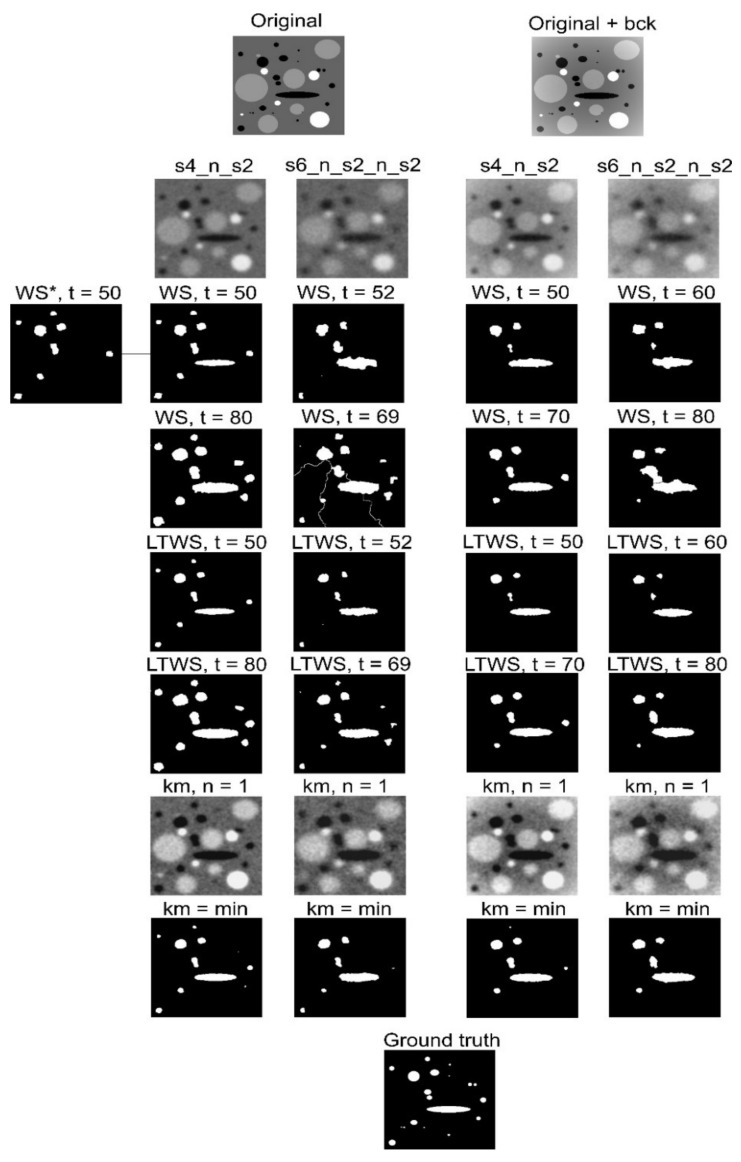
The results of pore segmentation test. Notations: bck—nonlinear background; s4_n_s2 and s6_n_s2_n_s2 denote the sequence of Gaussian blur (s) and Gaussian noise (n) added to the test image, see text for details; *t* denotes the threshold used for obtaining the marker image.

**Figure 3 jimaging-08-00242-f003:**
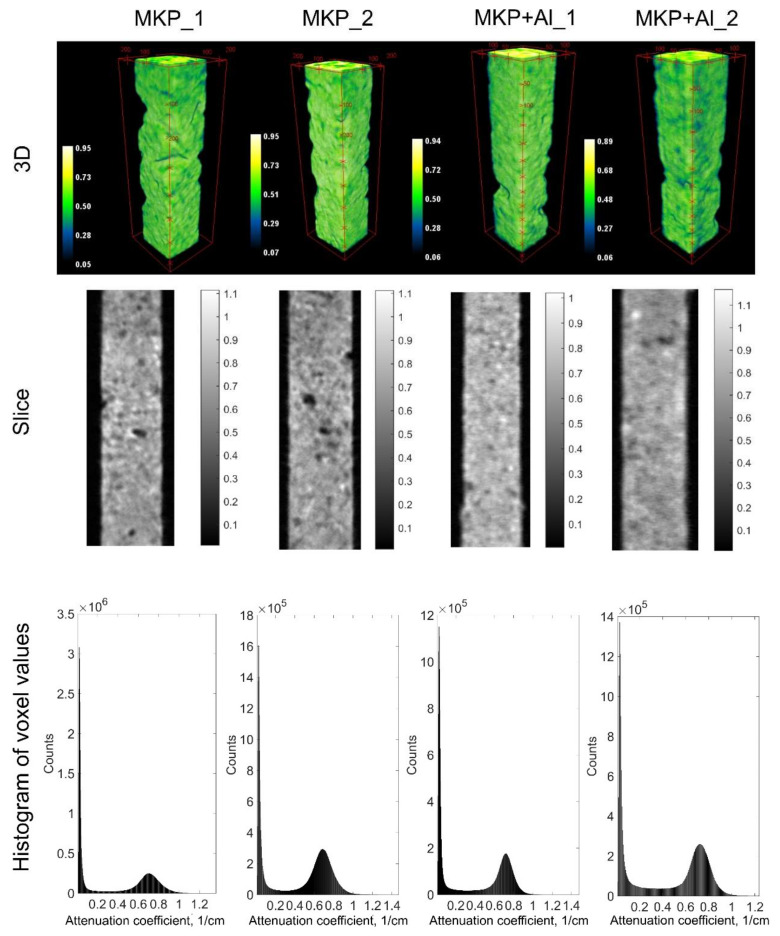
Results of the tomographic reconstruction of studied cement samples: 3D models, selected slices and the histograms of the neutron attenuation coefficient are shown. Color bars are presented in cm^−1^ units.

**Figure 4 jimaging-08-00242-f004:**
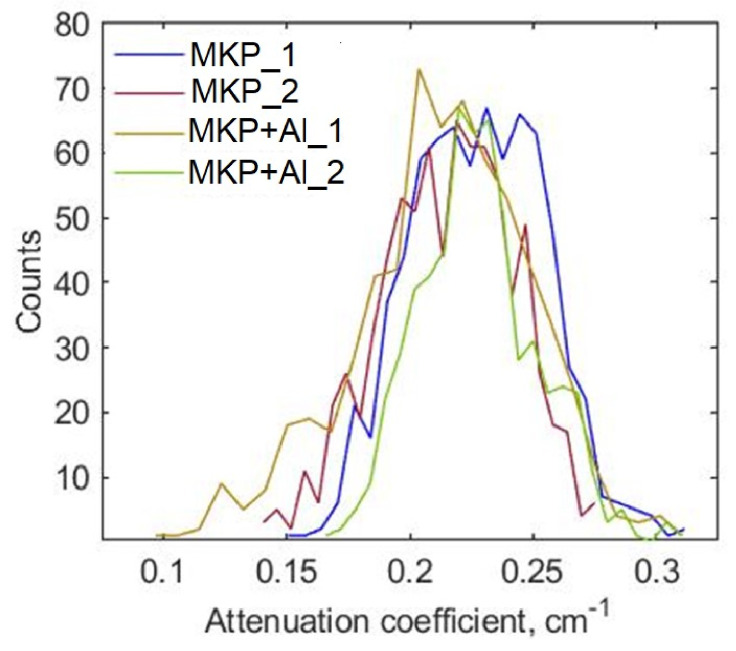
Distributions of the minimum gray values at the boundary between the samples and air calculated over the stack of the tomography slices.

**Figure 5 jimaging-08-00242-f005:**
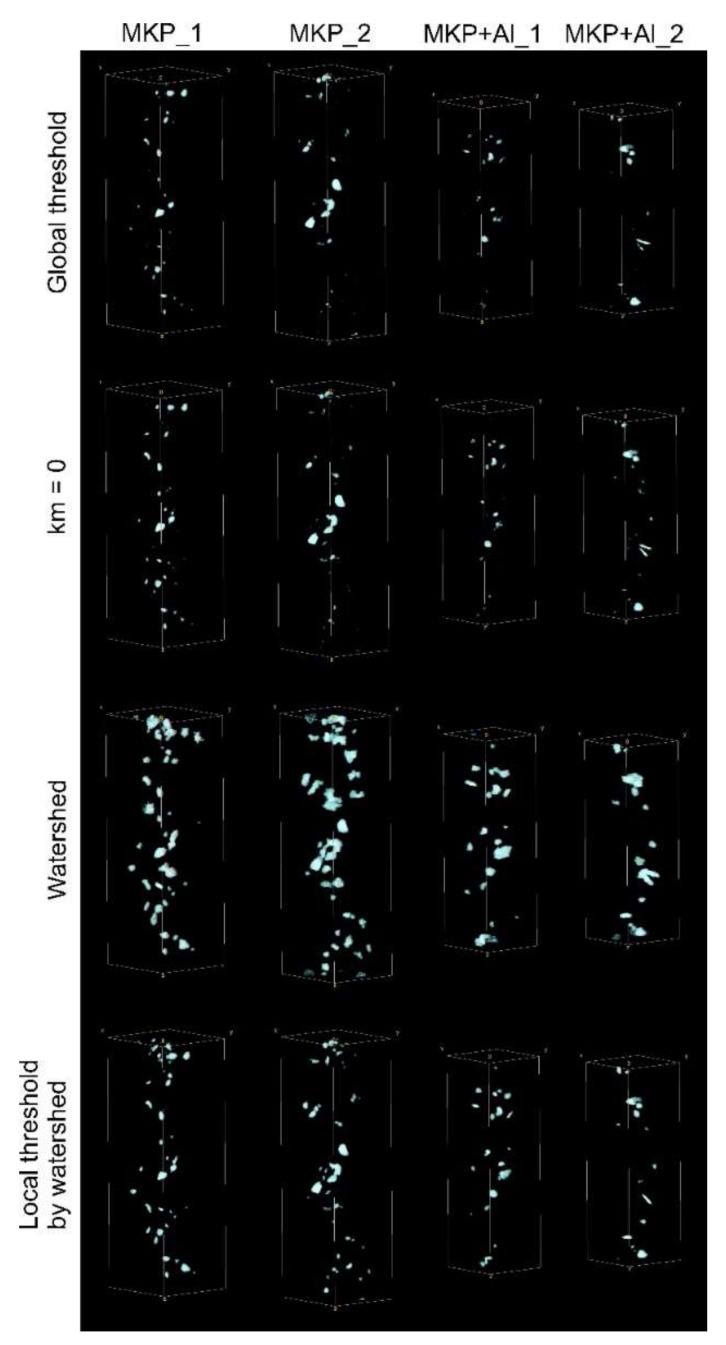
Three-dimensional models of spatial distribution of the segmented pores in the studied cement samples obtained by means of different segmentation techniques.

**Table 1 jimaging-08-00242-t001:** Jaccard index $J\left(ground\;truth,segmented\;image\right)$ calculated for the binary images of the segmented pores shown in Figure 2.

Method	s4_n_s2	Method	s6_n_s2_n_s2	Method	bck+s4_n_s2	Method	bck+s6_n_s2_n_s2
WS, t = 50	0.75	WS, t = 52	0.52	WS, t = 50	0.58	WS, t = 60	0.52
WS, t = 80	0.50	WS, t = 69	0.47	WS, t = 70	0.63	WS, t = 80	0.46
LTWS, t = 50	0.78	LTWS, t = 52	0.57	LTWS, t = 50	0.58	LTWS, t = 60	0.52
LTWS, t = 80	0.55	LTWS, t = 69	0.65	LTWS, t = 70	0.66	LTWS, t = 80	0.56
$\kappa^{m}\left(f\right)=0$	0.79	$\kappa^{m}\left(f\right)=0$	0.61	$\kappa^{m}\left(f\right)=0$	0.67	$\kappa^{m}\left(f\right)=0$	0.60

**Table 2 jimaging-08-00242-t002:** Samples porosities (%) calculated from the binary images shown in Figure 5.

Method	MKP_1	MKP_2	MKP+Al_1	MKP+Al_2
Global threshold	0.20	0.28	0.12	0.25
$\kappa^{m}\left(f\right)=0$	0.24	0.30	0.14	0.28
WS	1.33	1.71	1.11	1.23
LTWS	0.41	0.44	0.40	0.37

## Data Availability

The data presented in this study are available on request from the corresponding author.

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
