# Peer review of "Pore Segmentation Techniques for Low-Resolution Data: Application to the Neutron Tomography Data of Cement Materials"

_2313-433X, 2022, doi:10.3390/jimaging8090242_

Round 1

Reviewer 1 Report

I enjoyed reading this manuscript and I believe it will be an important contribution to pore segmentation. I would rank the importance of this manuscript as on par with the black body grid work of Lehmann and coworkers (ref 18).  However, I would like to see more detail in the algorithms and the experimental section.

Algorithms :

lines 120-130 The extended contrast enhance function is lightly discussed, with  a phrase "is constructed in the same way as equation 5". Come on, let's have a full discussion.

lines 165-169 The LTWS description is also incomplete.  Please, let's have more detail in the manuscript.

Experimental :

lines 232-246  Standard information that should be given include effective voxel size, length-over-pinhole diameter, sample to scintillator distance, type and thickness of scintillator, neutron spectrum (as pertinent, moderation, velocity selector, or monochromator).  Even better, please include image resolution as measured by a knife edge test at the sample position (not immediately in front of the scintillator).

lines 240-241 The experimental times are internally inconsistent; please double check or discuss.  That is, 20 seconds (line 240) x 360 projections (line 239) is 2 hrs, not "about 4 h" (line 241).  Were 2 frames collected with 20 second exposure and merged with zinger removal, making 40 seconds per projection? This is just a guess.

Fig 3 histograms.  Suggest using log counts so as to show more detail in the histogram.  Also, the label for the x-axis, "pixel value/cm-1" in not quite right. The numerical values look like "linear attenutation/cm-1".  The same labeling issue is present in Fig 4.

Fig 3 images are too small. The watershed lines are barely visible.

Equation 1 uses D' but the text line 52 uses D.

Author Response

Dear Editors and Reviewers,

We would like to re-submit our paper

“Pore segmentation techniques for the low-resolution data: application to the neutron tomography data of cement materials”

by Ivan Zel, Murat Kenessarin, Sergey Kichanov, Kuanysh Nazarov, Maria Bǎlǎșoiu, Denis Kozlenko

Ref: jimaging-1833127

I would like to sincerely thank Reviewers and Editors for careful reading of the manuscript and providing the useful remarks and comments.

We have made the following explanations and the corresponding corrections.

I enjoyed reading this manuscript and I believe it will be an important contribution to pore segmentation. I would rank the importance of this manuscript as on par with the black body grid work of Lehmann and coworkers (ref 18). However, I would like to see more detail in the algorithms and the experimental section.

Algorithms:

lines 120-130 The extended contrast enhance function is lightly discussed, with a phrase "is constructed in the same way as equation 5". Come on, let's have a full discussion.

lines 165-169 The LTWS description is also incomplete.  Please, let's have more detail in the manuscript.

We added the additional formulas and phrases for the extended contrast enhance function. We are not sure about what specifically should be discussed there. We have modified the text about LTWS just a bit. Upon personal request, we can provide the source script written in Matlab syntax. But we are not sure about the correctness of it licensing or copyright if we include it directly into the text.

Experimental:

lines 232-246  Standard information that should be given include effective voxel size, length-over-pinhole diameter, sample to scintillator distance, type and thickness of scintillator, neutron spectrum (as pertinent, moderation, velocity selector, or monochromator).  Even better, please include image resolution as measured by a knife edge test at the sample position (not immediately in front of the scintillator).

We used the geometry of the neutron beam as described in the publications Podurets, Crystallogr. Rep. 2021, 66(2), 254-266 and Kozlenko, Phys. Part. Nuclei Lett. 2016, 13, 346–351. Additional data about the features of the experiment like sample-to-scintillator distance were added to the section ‘4.1. Experimental’

lines 240-241 The experimental times are internally inconsistent; please double check or discuss.  That is, 20 seconds (line 240) x 360 projections (line 239) is 2 hrs, not "about 4 h" (line 241).  Were 2 frames collected with 20 second exposure and merged with zinger removal, making 40 seconds per projection? This is just a guess.

Indeed, the exposure time of one projection is 20 s, but it takes time to save the file, transfer it to the computer and rotation step of the goniometer. All these additional operations take an additional 20 seconds to the exposure itself. In order to remove the discrepancy and raise additional questions, we have corrected the sentence about exposition time.

Fig 3 histograms.  Suggest using log counts so as to show more detail in the histogram.  Also, the label for the x-axis, "pixel value/cm-1" in not quite right. The numerical values look like "linear attenutation/cm-1".  The same labeling issue is present in Fig 4.

We have corrected the figures. We did not use log counts for better perception of histogram features (peaks and valleys).

Fig 3 images are too small. The watershed lines are barely visible.

In any case, the Figure 3 will be reduced accordingly when published. If an HD version of the drawing needed, the reader can download it online from the journal page or we can provide them.

Equation 1 uses D' but the text line 52 uses D.

This is a comma after the formula, not an apostrophe. We fixed it.

Sincerely,

Zel Ivan and Kichanov Sergey,

behalf of co-authors.

Reviewer 2 Report

Dear Authors,
congratulations for the paper but I have a remark. The experimental part supporting the pore segmentation technique seems to be limited and should be improved.
It would have been better to try this technique with at least one other type of cement and / or material with higher porosity.

Author Response

Dear Editors and Reviewers,

We would like to re-submit our paper

“Pore segmentation techniques for the low-resolution data: application to the neutron tomography data of cement materials”

by Ivan Zel, Murat Kenessarin, Sergey Kichanov, Kuanysh Nazarov, Maria Bǎlǎșoiu, Denis Kozlenko

Ref: jimaging-1833127

I would like to sincerely thank Reviewers and Editors for careful reading of the manuscript and providing the useful remarks and comments.

We have made the following explanations and the corresponding corrections.

congratulations for the paper but I have a remark. The experimental part supporting the pore segmentation technique seems to be limited and should be improved.

In accordance with your comment and the comments of other reviewers, we have expanded the description of the proposed methods.

It would have been better to try this technique with at least one other type of cement and / or material with higher porosity.

We agree. We believe that this is a universal method, and cement materials were used as an example. However, in cement materials porosity is not very high at coarse scale. Unfortunately, we have not other porous materials to measure now. We plan to measure other cement samples in the nearest future and use the presented approaches for pore segmentation.

Sincerely,

Zel Ivan and Kichanov Sergey,

behalf of co-authors.

Reviewer 3 Report

This manuscript has presented an application of neutron imaging on cement material. The new techniques for the pore segmentation in the low-resolution images or tomography data have been presented by using global thresholding of the enhanced contrast data and local threshold by the watershed (LTWS). The performed tests have demonstrated their their advantages over the conventional marker-based watershed technique. The considered techniques were applied to the neutron tomography data of the MKP cement samples. The following comparison of the segmented data as well as of the calculated porosity of the cement samples confirmed the results of tests, showing the failing of the conventional watershed as compared to the proposed techniques.

Some questions are listed in the following:

1.        At first, the facility for this research work should be introduced a bit, otherwise the reader doesn’t know exactly what kind of neutron source can be used to do such work. What is the lever of neutron intensity, please give some reference for source introduction, and including the low resolution level for the neutron radiography or tomography.

2.        Formula (1) should be written as dh>=l  . 1/(L/D) , D shouldn’t has the sign of ‘

3.        Line 112, ‘trinarizaiton’ should be ‘trinarization’

4.        Fig.5.3, one of ‘different’ should be deleted.

5.        Quantitative assessment of the difference between the ground truth and the segmented images was performed using the Jaccard index (see Table 1) defined as ?(?, ?) =|??|/|??|. I suggest the authors list the reference to introduce the Jaccard index, describes more its meaning and explain how important it is to evaluate your index in table 1 or to porosity measurements.

Author Response

Dear Editors and Reviewers,

We would like to re-submit our paper

“Pore segmentation techniques for the low-resolution data: application to the neutron tomography data of cement materials”

by Ivan Zel, Murat Kenessarin, Sergey Kichanov, Kuanysh Nazarov, Maria Bǎlǎșoiu, Denis Kozlenko

Ref: jimaging-1833127

I would like to sincerely thank Reviewers and Editors for careful reading of the manuscript and providing the useful remarks and comments.

We have made the following explanations and the corresponding corrections.

This manuscript has presented an application of neutron imaging on cement material. The new techniques for the pore segmentation in the low-resolution images or tomography data have been presented by using global thresholding of the enhanced contrast data and local threshold by the watershed (LTWS). The performed tests have demonstrated their their advantages over the conventional marker-based watershed technique. The considered techniques were applied to the neutron tomography data of the MKP cement samples. The following comparison of the segmented data as well as of the calculated porosity of the cement samples confirmed the results of tests, showing the failing of the conventional watershed as compared to the proposed techniques.

Some questions are listed in the following:

At first, the facility for this research work should be introduced a bit, otherwise the reader doesn’t know exactly what kind of neutron source can be used to do such work. What is the lever of neutron intensity, please give some reference for source introduction, and including the low resolution level for the neutron radiography or tomography.

We used the geometry of the neutron beam as described in the publications Podurets, Crystallogr. Rep. 2021, 66(2), 254-266 and Kozlenko, Phys. Part. Nuclei Lett. 2016, 13, 346–351. Additional data about the features of the experiment like sample-to-scintillator distance or voxel size were added to the section ‘4.1. Experimental’

Formula (1) should be written as dh>=l  . 1/(L/D) , D shouldn’t has the sign of

This is a comma after the formula, not an apostrophe. We fixed it.

Line 112, ‘trinarizaiton’ should be ‘trinarization’

Thank you. We corrected it.

Fig.5.3, one of ‘different’ should be deleted.

We check text and perform a lot of corrections. Thank you again.

Quantitative assessment of the difference between the ground truth and the segmented images was performed using the Jaccard index (see Table 1) defined as ?(?, ?) =|?∩?|/|?∪?|. I suggest the authors list the reference to introduce the Jaccard index, describes more its meaning and explain how important it is to evaluate your index in table 1 or to porosity measurements.

We added the reference for the Jaccard index and some additional phrases.

Sincerely,

Zel Ivan and Kichanov Sergey,

behalf of co-authors.